# PTA-Sync: Packet-Train-Aided Time Synchronization for Underwater Acoustic Applications

A-Ra Cho and Youngchol Choi *

Ocean and Maritime Digital Technology Research Division, Korea Research Institute of Ship & Ocean Engineering (KRISO), Daejeon 34103, Republic of Korea; zoazoa@kriso.re.kr
* Correspondence: ycchoi@kriso.re.kr; Tel.: +82-42-866-3837

**Abstract:** In underwater acoustic applications, time synchronization errors accumulate severely with time, owing to the unpredictable propagation delay change induced by mobility and low sound propagation speed. In previously reported schemes, error accumulation during underwater acoustic time synchronization occurred, decreasing their performance and affecting their applicability. To overcome this limitation, we propose packet-train-aided time synchronization (PTA-Sync) for underwater acoustic application. The proposed PTA-Sync adaptively tracks one-way travel time (OWTT) change using three-dimensional linear velocity vector, the transmit time difference and arrival time difference of two adjacent packets in a packet train, and the position information of a mobile reference node. Thus, this scheme enables us to reduce the synchronization error accumulation rate by estimating the propagation delay change more accurately. Simulation results show PTA-Sync achieves higher accuracy than existing OWTT-based synchronization schemes. Thus, PTA-Sync can be effectively used in underwater exploratory activities because it can successfully reduce time synchronization errors.

**Keywords:** underwater acoustic time synchronization; one-way message exchange; reception time difference





## 1. Introduction

Underwater exploratory activities, such as underwater terrain exploration, seismic monitoring, maritime defense, and marine environment and ecosystem monitoring, require simultaneous procurement of data of a wide area in the ocean, thus necessitating the need for an underwater communication network that can facilitate effective communication between multiple nodes. In underwater acoustic networks, time synchronization (TS) is crucial for maintaining concurrency in the data; moreover, TS is mandatory in a network protocol that operates based on time information. For example, TS should be present between the nodes in accurate data transmission to allocated time slot in time division multiple access (TDMA) as well as in the back-off algorithm, which is widely used in IEEE 802.11 media access control (MAC) protocol [1].

The TS protocol of a terrestrial network, which employs radio waves, can be classified into three categories: 1. single hop vs. multiple hop; 2. stationary node vs. mobile node; and 3. MAC layer based vs. standard method [2–4]. The most commonly used TS protocols of terrestrial networks are reference broadcast synchronization (RBS) [5], timing-sync protocol for sensor networks (TPSN) [6], flooding time synchronization protocol (FTSP) [7], and tree-based synchronization algorithm [8]. Koeptz et al. [9] and Horauer et al. [10] have analyzed TS error elements of a terrestrial network and classified them into fixed delay (transmission time, propagation time, and reception time) and random delay (send time, access time, and receive time) elements. The uncertainty of fixed delay is taken care of by using statistical TS algorithms in [11,12]. These TS protocols of terrestrial networks cannot be directly applied to underwater networks because acoustic waves, rather than radio waves, are used under water; radio waves are extremely attenuated in water. Underwater

acoustic time synchronization protocols (UATSPs), which employ acoustic waves, need to consider several unique disadvantages, such as inability to use a global positioning system (GPS), slow propagation speed, narrow bandwidth due to physical limitations of acoustic sensors, rapid changing of channels over time, and high mobility of nodes due to ocean currents [13–16]. In particular, the propagation time delay caused by the underwater acoustic wave speed, which is approximately 200,000 times lower than that of the radio wave, is the most important constraint in TS design. The propagation time delay varies depending on the acoustic speed and the node's mobility. Acoustic speed is a spatial–temporal variable as it varies depending on the depth of the water owing to the influence of temperature, pressure, and salinity, and it is difficult to predict the node's mobility because of ocean currents that change depending on the surrounding environment. The performance of UATSP is substantially affected by the estimation accuracy of propagation delay; therefore, for improving the estimation accuracy of the propagation delay, the effects of acoustic wave speed and mobility must be considered.

TSHL [17], which is the first UATSP that considers the underwater propagation delay, was proposed for stationary nodes. MU-Sync [18] is a cluster-based UATSP that considers the mobility of nodes; it improved the synchronization performance by calculating the half of round-trip time (RTT) as propagation delay. To improve the accuracy and energy efficiency, Mobi-Sync [19] uses a spatial correlation of node speeds for estimating the changes in the propagation delay. D-Sync [20], DA-Sync [21], TSMU [22], and DE-Sync [23] are UATSPs that utilize Doppler shift caused by the relative mobility of nodes to improve the synchronization accuracy and energy efficiency. APE-Sync [24] aims to achieve a high-precision and energy efficiency by combining DE-Sync and the Kalman filter tracking time-varying clock skew. PCDE-Sync [25] combines with partial clustering method built on the artificial fish swarm algorithm and clock correction method by the Doppler shift, and it achieves a trade-off of accuracy and energy consumption. Joint time synchronization and localization (JSL) [26] compensates the stratification effect caused by the inhomogeneity of underwater media, and improves the accuracy by combining the localization and TS. Robust joint localization and synchronization (RJLS) [27] improves the synchronization precision by repeatedly performing calculations until clock skew and offset converge after constant application of the linear least squares method to the method combining synchronization and localization. Such UATSPs are two-way travel time (TWTT)-based synchronization schemes wherein a packet for synchronization is sent from a source node to a receiver node, which sends back a response packet to the source node; subsequently, the two-way travel time is measured. This scheme has the advantage of the delay time being calculated using only the reference clock information of the source node, without using the clock information of the receiver node. However, the RTT of the packet is considerably high in an environment with propagation delays, e.g., an underwater environment, thus increasing the uncertainty of the propagation delay due to the movements of nodes, ultimately leading to errors in the propagation delay estimation.

In contrast, in one-way travel time (OWTT)-based synchronization schemes, the propagation delay is measured based on the time required to send a packet from the source node to the receiver node; these schemes have been frequently applied in studies related to navigation [28,29]. The OWTT schemes calculate propagation delay by using the clock information of the receiver node and the reference clock of the source node; therefore, propagation delay measurements may vary depending on the synchronization accuracy of the receiver node clocks. However, as multiple nodes simultaneously receive beacon packets to initiate synchronization, the overhead of the entire network can be reduced, thereby lowering the packet reception time by more than 50% compared to that of the TWTT schemes. Because low frequency acoustic communication in bandwidths of several kHz to some tens of kHz is primarily used in underwater environments, message transmission is limited owing to large propagation and transmission delays. Therefore, the OWTT-based synchronization is more advantageous than TWTT-based synchronization in underwater acoustic applications, wherein minimization of message overhead and delay is crucial.

In [29], TS and localization were achieved by conducting a sea experiment on an unmanned underwater vehicle (UUV) using the one-way ranging estimation technique. Liu et al. [30] also proposed an energy-efficient OWTT scheme based on nonlinear clock skew tracking in the clock skew and non-Gaussian noises by applying the Gaussian mixture model. In E2DTS [31], the influence of changes in the propagation delay was minimized using the transmission and reception time difference of continuous packets. However, there was a drawback: the changes in OWTT with respect to the moving speed of the node were assumed to be consistent. TSMA-IGTS [32] improved the synchronization accuracy by estimating the changes in OWTT based on the estimation of relative speed through the Doppler effect in an underwater acoustic network. In this way, the estimated changes in OWTT were deduced from the relative speed, rather than velocity; therefore, the estimated changes were accurate only when the movement directions of the reference node and ordinary nodes were identical, and they were considered inaccurate if the directions were unidentical.

This study proposes a packet-train-aided time synchronization (PTA-Sync) protocol of packet-train communication, which can reduce TS errors by tracking the changes in the propagation delay caused by node movements and slow propagation speeds in an underwater acoustic network environment. PTA-Sync adopts OWTT for tracking propagation delay changes and propagation delay asymmetry caused by mobility using the measured linear speed. In particular, synchronization is performed using the position information of the reference node, the measured three-dimensional speed vector of the node, the time difference in the transmitted packets, and the time difference in the received packets. This enables continuous estimation of changes in the reference node and propagation delay for each packet section during the process of continuously transmitting a beacon as several sub-frames in the form of a packet train. Unlike the conventional OWTT-based synchronization method, the TS accuracy is increased by estimating the propagation delay change by applying three-dimensional vector values instead of scalar quantities to the node's moving speed. As a result of performing simulations according to various network condition changes, such as the synchronization elapsed time, node's average velocity, number of synchronization message exchanges, synchronization message transmission time interval, the degree of mobility randomness, and network range, it is observed that the time error synchronization performance of PTA-Synch is improved by 87.83% and 62.43% compared to that of TSMA-IGTS and E2DTS, respectively.

The following parts of the paper are organized as follows. Section 2 elaborates on the target network and the proposed PTA-Sync protocol. Section 3 comparatively analyzes the synchronization performance of the PTA-Sync protocol against TSMA-IGTS andE2DTS. Finally, Section 4 presents the concluding points of the study.

## 2. Synchronization Protocol for UANets

### 2.1. Underwater Acoustic Networks (UANets)

Figure 1 shows the target cluster-oriented underwater acoustic network model, which comprises one command ship (CS) and multiple underwater nodes (UN). CS is positioned on the ocean surface and moves along the ocean current; it controls the entire network. CS is connected to a GPS and performs clock synchronization, thus functioning as a reference clock for UN. CS is equipped with an underwater acoustic communication modem (UAC) for communicating with UN, and periodically broadcasts a beacon containing transmission time information to UN for synchronization.

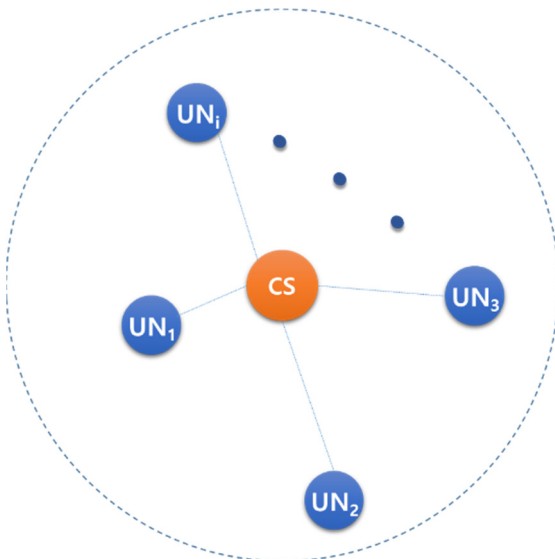

**Figure 1.** Target network model.

UN, which is a mobile node similar to an autonomous underwater vehicle (AUV) or a submarine, is self-propelled. UN collects underwater data or performs underwater missions delivered from CS. All UNs are within the acoustic communication range of CS and communicate with CS. UN renews the position information by obtaining its own moving speed and moving direction information through mounted sensors. UN synchronizes the CS and UN clock times by using the transmission and reception times of CS that have been periodically received.

### 2.2. PTA-Sync Protocol

Assumption and Overview of PTA-Sync

We make the following assumptions:

1.    UN contains the initial position information and initial propagation delay information.
2.    UN acquires the position and speed from sensors, which include an altitude heading reference system (AHRS) and a Doppler velocity log (DVL).

Table 1 defines the parameters related to the PTA-Sync results.

**Table 1.** Parameter description of PTA-Sync.

| Parameter | Description |
|:---:|:---:|
| $N$ | Total number of sub-frame beacons |
| $B_j$ | $j$-th sub-frame beacon |
| $t(j)$ | Time at which CS transmitted $B_j$ |
| $RT(j)$ | Time at which UN received $B_j$ |
| $D(j)$ | One-way travel time (OWTT) for $B_j$ transmitted by CS to be received by UN |
| $\boldsymbol{P}_{CS}(j)$ | 3D position vector when CS transmits $B_j$ |
| $\boldsymbol{P}_{UN}(j)$ | 3D position vector when UN receives $B_j$ |
| $\boldsymbol{V}_{UN}(j)$ | 3D velocity vector when UN receives $B_j$ |
| $\Delta D(j)$ | Changes in OWTT of UN when $B_{j-1}$ and $B_j$ are transmitted |
| $\hat{a}(j)$ | Estimated clock skew of UN when $B_{j-1}$ and $B_j$ sare transmitted |
| $\bar{a}$ | Mean of $\hat{a}$ |
| $\hat{b}$ | Estimated clock offset of UN |

Figure 2 shows the parameters applied to PTA-Sync of CS and UN, and elucidates the parameters defined in Table 1. A beacon broadcast by CS comprises *N* number of sub-

frames, and is transmitted in the form of a packet train. Each sub-frame beacon contains the position information of CS and time stamp transmission time $t(j)$.

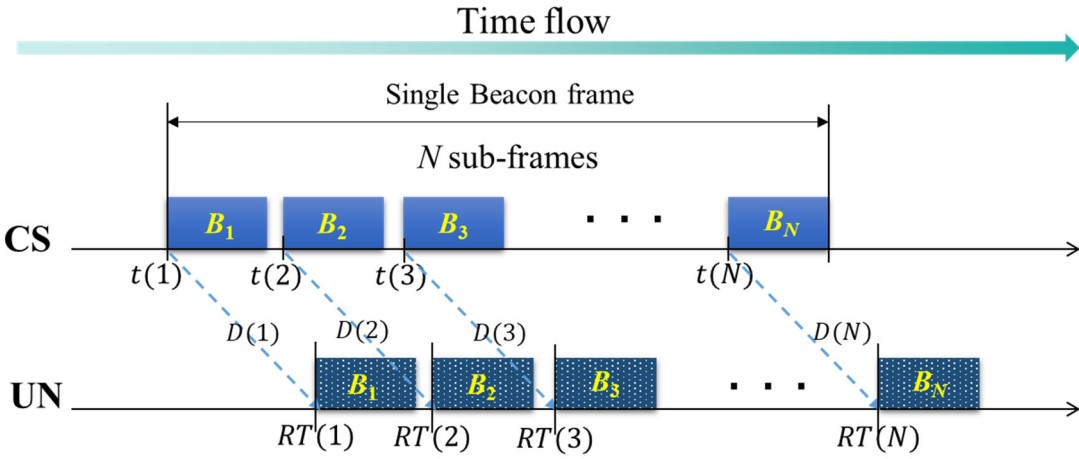

**Figure 2.** Illustration of parameters for PTA-Sync.

Description of PTA-Sync

The relationship between the reference time of CS and local time of UN can be expressed using the following linear function:

$$RT(j) = a \cdot (t(j) + D(j)) + b, \tag{1}$$

where $a$ is the clock skew of UN, and $b$ is the clock offset of UN.

The transmission time difference between $B_{j-1}$ and $B_j$ in CS, $\Delta t(j)$, can be expressed as follows:

$$\Delta t(j) = t(j) - t(j-1). \tag{2}$$

The reception time difference between $B_{j-1}$ and $B_j$ in UN, $\Delta RT(j)$, can be expressed as follows:

$$\Delta RT(j) = RT(j) - RT(j-1). \tag{3}$$

Equation (4) can be expressed as follows using Equations (1)–(3):

$$\Delta RT(j) = a \cdot (\Delta t(j) + \Delta D(j)), \tag{4}$$

where $\Delta D(j)$ is defined as follows:

$$\Delta D(j) = D(j) - D(j-1). \tag{5}$$

To estimate $D(j)$, $\boldsymbol{P}_{UN}(j)$ must be determined. When UN receives $B_j$, the changes in the position of UN are calculated using the speed and direction information at the reception point of $B_{j-1}$ to calculate the propagation delay changes during $\Delta t(j)$.

$$\boldsymbol{P}_{UN}(j) = \boldsymbol{P}_{UN}(j-1) + \boldsymbol{V}_{UN}(j-1) \cdot \Delta t(j); \tag{6}$$

note that $\boldsymbol{P}_{UN}$ and $\boldsymbol{V}_{UN}$ are three-dimensional vectors.

UN estimates the propagation delay between CS and UN using the information $P_{CS}(j)$ obtained by receiving $P_{UN}(j)$ and $B_j$, and can be expressed as follows:

$$\hat{D}(j) = |P_{CS}(j) - P_{UN}(j)|/c, \tag{7}$$

where $c$ is the underwater acoustic speed.

Using Equations (4), (5) and (7), the clock skew of UN, $\hat{a}(j)$, can be estimated while $B_{j-1}$ and $B_j$ are being transmitted.

$$\hat{a}(j) = \frac{\Delta RT(j)}{\Delta t(j) + \Delta D(j)}. \tag{8}$$

The mean clock skew of UN, $\bar{a}$, can be determined by obtaining the average of $\hat{a}(j)$ as follows:

$$\bar{a} = \frac{1}{N-1} \sum_{j=2}^{N} \hat{a}(j). \tag{9}$$

The clock offset of UN, $\hat{b}$, can be determined using $\bar{a}$ and Equation (1) as follows:

$$\hat{b} = RT(1) - \bar{a} \cdot [t(1) + D(1)]. \tag{10}$$

The entire operation procedure of PTA-Sync can be expressed as the following pseudo code (Algorithm 1):

---
**Algorithm 1:** PTA-Sync

---
1:　**Get** $P_{UN}(1)$
2:　**Get** $RT(j)$ and $V_{UN}(j)$, $1 \le j \le N$
3:　**Get** $P_{CS}(j)$ and $t(j)$ by parsing $B_j$, $1 \le j \le N$
4:　**Set** $j = 1$,
5:　**while** $j \le N$ **do**
6:　　　**if** $j > 1$
7:　　　　　$\Delta RT(j) = RT(j) - RT(j-1)$
8:　　　　　$\Delta t(j) = t(j) - t(j-1)$
9:　　　　　$P_{UN}(j) = P_{UN}(j-1) + V_{UN}(j-1) \cdot \Delta t(j)$
10:　　　　$\hat{D}(j) = |P_{CS}(j) - P_{UN}(j)|/c$　　　　　　// c is the underwater acoustic speed
11:　　　　$\Delta D(j) = D(j) - D(j-1)$
12:　　　　$\hat{a}(j) = \frac{\Delta RT(j)}{\Delta t(j) + \Delta D(j)}$
13:　　　**end if**
14:　　　$j++$
15:　**end while**
16:　$\bar{a} = \text{mean}\{\hat{a}\}$
17:　$\hat{b} = RT(1) - \bar{a} \cdot [t(1) + D(1)]$

---

## 3. Simulation and Results

The simulation results of PTA-Sync are analyzed in this section. For analyzing the performance of PTA-Sync for UANets, the synchronization performance of PTA-Sync was compared with that of E2DTS [31] and TSMA-IGTS [32], which are OWTT-based synchronization schemes. The synchronization performance metric of these protocols is the time error between CS and UN after performing synchronization where the synchronization performance is compared by changing the time elapsed after synchronization ($T$), changes in mean velocity of UN ($V\_mean$), number of messages exchanged for synchronization ($N\_ex$), synchronization message transmission time interval ($I\_tx$), network range ($Nr$), and the degree of mobility randomness ($\alpha$).

### 3.1. Simulation Setup

UN is a self-propelled node that uses the Gauss–Markov model [33], in which the mobility is modeled as follows:

$$S_j = \alpha S_{j-1} + (1 - \alpha)\overline{S} + \sqrt{(1 - \alpha^2)}S^{G_{j-1}}, \tag{11}$$

$$R_j = \alpha R_{j-1} + (1 - \alpha)\overline{R} + \sqrt{(1 - \alpha^2)}R^{G_{j-1}}. \tag{12}$$

Here, $S_j$ and $R_j$ are the three-dimensional velocity vector and direction when $B_j$ is transmitted; $\alpha$ is the degree of mobility randomness and is in the range of $0 \le \alpha \le 1$. As the degree of mobility randomness increases, the value of $\alpha$ decreases from 1 to 0. In other words, as $\alpha$ approaches 0, UN has a movement independent of the previous movement, whereas as $\alpha$ approaches 1, it is affected more by the previous movement pattern's velocity and direction. $\overline{S}$ and $\overline{R}$ are the mean velocity and mean direction of UN which are constant; and $S^{G_{j-1}}$ and $R^{G_{j-1}}$ are random variables that have Gaussian distributions.

The simulation conditions are as follows:

1.　The initial position of UN has a uniform distribution in [0, Nr], and the initial propagation delay of UN is known.
2.　UN maintains its velocity and direction during one sub-frame.
3.　UN's propulsion velocity and direction are distorted by the underwater environment, and the distorted velocity and direction have Gaussian distributions, with a mean of 0 and a standard deviation of $\sqrt{(1 - \alpha^2)}$.
4.　The acoustic speed is constant at 1500 m/s.
5.　All frame data are transmitted successfully without loss or error.
6.　The data transmission rate is 100 bps, and the packet length is variable.
7.　The arbitrary mobility level in the Gauss–Markov model is 0.5.
8.　The simulation is repeated 20,000 times.

Table 2 shows the parameters applied to the simulation. The parameters are set based on the previous related works [17–31,34].

**Table 2.** Simulation parameters.

| Parameter | Value |
|---|---|
| Mean velocity of UN | [0.5, 5] m/s |
| Number of messages exchanged for synchronization | [1, 30] |
| Network range | [500, 30,000] m |
| Clock skew | Uniform distribution within the range of 20–50 ppm. |
| Degree of mobility randomness | [0, 1] in the Gauss–Markov model |
| Mean direction of UN | $\pi$ |
| Data transmission rate | 100 bps |
| Packet size | Varying in the range of [1, 750] bytes |

### 3.2. Simulation Results

Figure 3 shows the time error according to the elapsed time after synchronization. In this simulation, V_mean is 2.5 m/s and 0.5 m/s, I_tx is 1.2 s, N_ex is 10, Nr is 15 km, and $\alpha$ is 0.5. As shown in Figure 3, as T increases, the time error of all synchronization protocols linearly increases, but the increase rates vary. The time errors of TSMA-IGTS and E2DTS increase at a higher rate than that of PTS-Sync owing to the OWTT estimation method. TSMA-IGTS and E2DTS calculate the changes in OWTT by considering the scalar speed rather than vector velocity of UN, thus resulting in errors between the actual OWTT changes and propagation OWTT change estimates from the assumption that moving directions of

CS and UN are identical. Therefore, TSMA-IGTS and E2DTS have increasing time errors because propagation delay estimation errors abruptly increase over time in relation to PTA-Sync. Ultimately, PTA-Sync demonstrated the highest synchronization performance in all three protocols as the time elapsed.

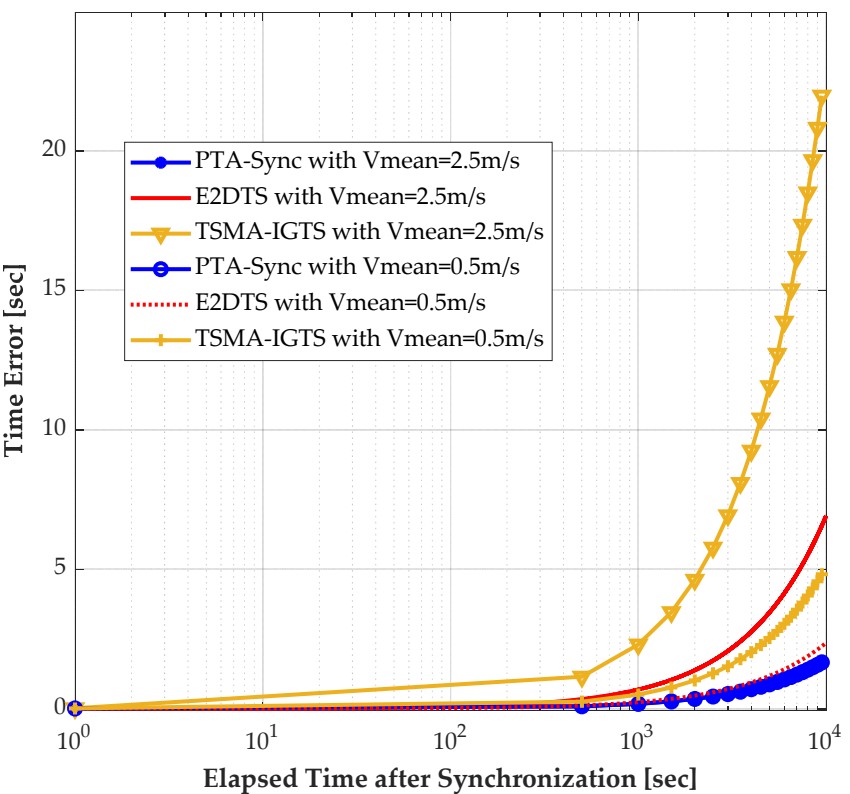

**Figure 3.** Time error measured at 30 s after synchronization versus $T$ when $V\_mean$ = 2.5 m/s, 0.5 m/s, $I\_tx$ = 1.2 s, $N\_ex$ = 30, $Nr$ = 15 km, $\alpha$ = 0.5.

Figure 4 shows the time error of synchronization protocols with respect to the changes in the mean velocity of UN. As $V\_mean$ increases, the time error of PTA-Sync remains relatively consistent, while those of TSMA-IGTS and E2DTS increase. PTA-Sync is not affected by the increase in velocity because the vector quantity is adequately tracked even when the velocity increases. However, TSMA-IGTS and E2DTS have larger time errors as propagation delay fluctuation increases when the speed increases due to discrepancy between the vector velocity and scalar speed of UN.

Figure 5 shows the synchronization time error with respect to the number of messages sent for synchronization when $V\_mean$ = 2.5 m/s, $I\_tx$ = 1.2 s, $Nr$ = 15 km, $T$ = 30 s, and $\alpha$ = 0.5. The time error of all three synchronization protocols decreases as $N\_ex$ increases, but the time error becomes saturated at a specific value of $N\_ex$. As the number of messages sent for synchronization increases, the amount of time information also increases, thus increasing synchronization accuracy. However, if $N\_ex$ increases beyond a certain point, there is a limitation in performance improvement because synchronization performance is determined mostly by the Gauss probability variables in Equations (11) and (12). PTA-Sync demonstrated 87.83% and 62.43% performance improvement compared to TSMA-IGTS and E2DTS, respectively, when $N\_ex$ = 15, $V\_mean$ = 2.5 m/s, and $Nr$ = 15 km. Although the performance improvement in Figure 5b is relatively lower than that of Figure 5a, PTA-Sync still has the best performance in all three protocols. The complexity of PTA-Sync is $O(N\_ex)$ which is the same as those of TSMA-IGTS and E2DTS, because all three protocols employ one-way TS packet transmission in synchronization process.

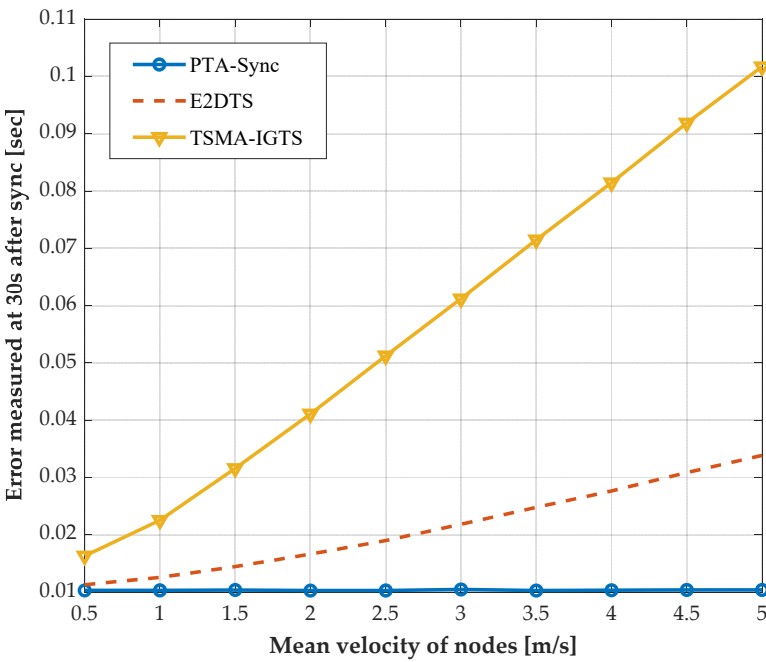

**Figure 4.** Time error measured at 30 s after synchronization versus *V_mean* when *I_tx* = 1.2 s, *N_ex* = 30, *Nr* = 15 km, *T* = 30 s, *α*=0.5.

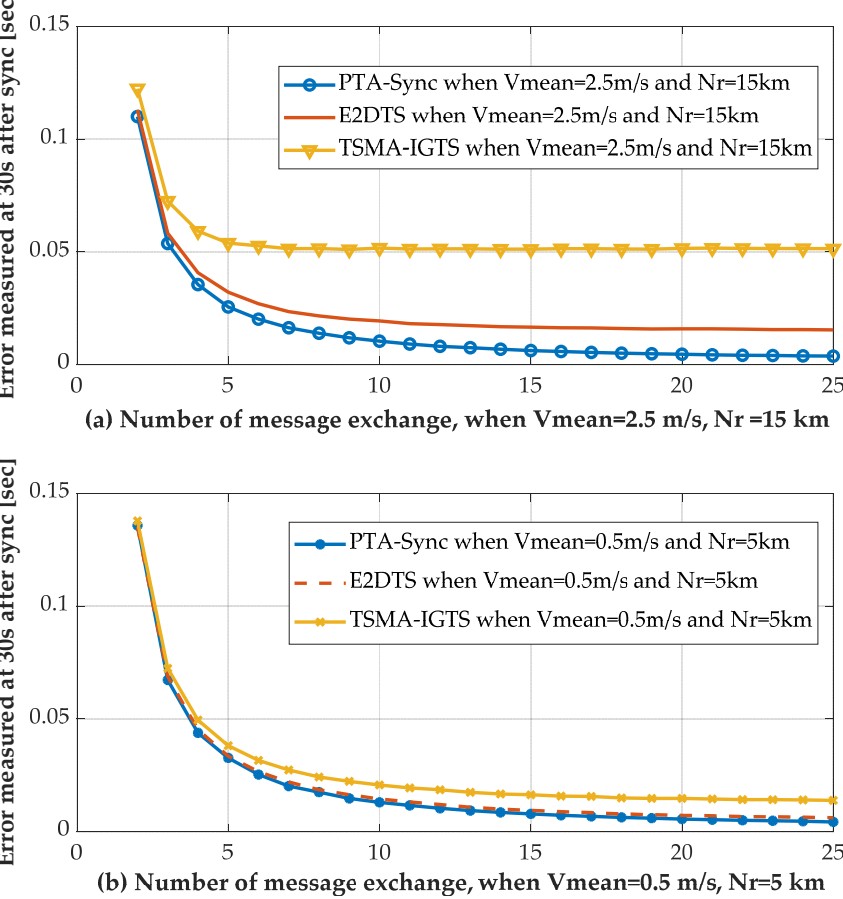

**Figure 5.** Time error measured at 30 s after synchronization versus *N_ex* when *I_tx* = 1.2 s, *T* = 30 s, *α* = 0.5.

Figure 6 shows the synchronization performance with respect to the message transmission time interval for synchronization when $N\_ex$ = 10, $V\_mean$ = 2.5 m/s, $Nr$ = 1.5 km, $T$ = 30 s, and $\alpha$ = 0.5. As shown in the figure, the time error decreases as $I\_tx$ increases and then becomes saturated at a specific value of $I\_tx$. Propagation delay changes increase as $I\_tx$ increases, while transmission time difference and reception time difference also increase. $\Delta D$ is smaller than the increase in $\Delta t$ and $\Delta RT$. Consequently, as $I\_tx$ increases, the synchronization time error becomes saturated when $\frac{\Delta RT}{\Delta t}$ becomes a dominant factor in Equation (8). In Figure 6a, PTA-Sync improves the synchronization performance by 81.78% and 60.99% on an average in relation to TSMA-IGTS and E2DTS, respectively.

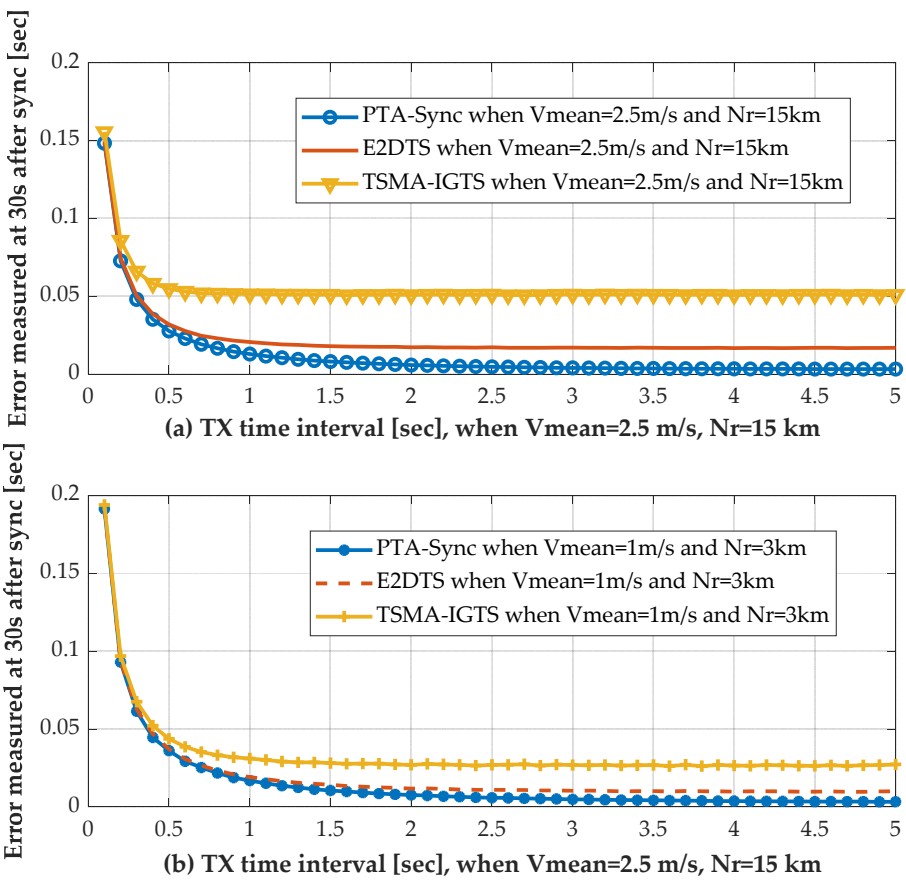

**Figure 6.** Time error measured at 30 s after synchronization versus $I\_tx$ when $N\_ex$ = 10, $T$ = 30 s, $\alpha$ = 0.5.

Figure 7 shows the synchronization performance according to the degree of mobility randomness when $V\_mean$ = 1 m/s, 2.5 m/s, 5 m/s, $N\_ex$ = 10, $I\_tx$ = 1.2 s, $Nr$ = 15 km, and $T$ = 30 s. As $\alpha$ increases, the node has more similarity to the movement pattern of the previous sub-frame and moves more smoothly. Based on the simulation assumptions, the node's movement distortion has a Gaussian distribution, with a mean of 0 and a standard deviation of $\sqrt{(1-\alpha^2)}$, and as $\alpha$ increases, the UN's movement distortion decreases. Accordingly, PTA-Sync shows that as $\alpha$ increases, the movement distortion decreases, which reduces the propagation delay estimation error, thus reducing the synchronization time error. In contrast, E2DTS and TSMA-IGTS show that as $\alpha$ increases, the synchronization time error increases. Because of the mismatch between the UN's vector velocity and scalar speed, as $\alpha$ increases when the movement direction of UN and CS match, the propagation delay estimation error decreases. However, when the movement directions do not match, although there is a similarity to the movement pattern of the previous frame, the propagation delay estimation error increases. Therefore, when $0 \leq \alpha \leq 0.5$, the similarity of the movement pattern with the previous frame is low, and the influence

of the mobility randomness is dominant. Thus, the propagation delay estimation error is constant, regardless of $\alpha$, and the synchronization time error has a nearly constant value. When $0.5 \leq \alpha \leq 1$, as the similarity to the previous frame's movement pattern increases, the propagation delay estimation error increases, thus increasing the synchronization time error.

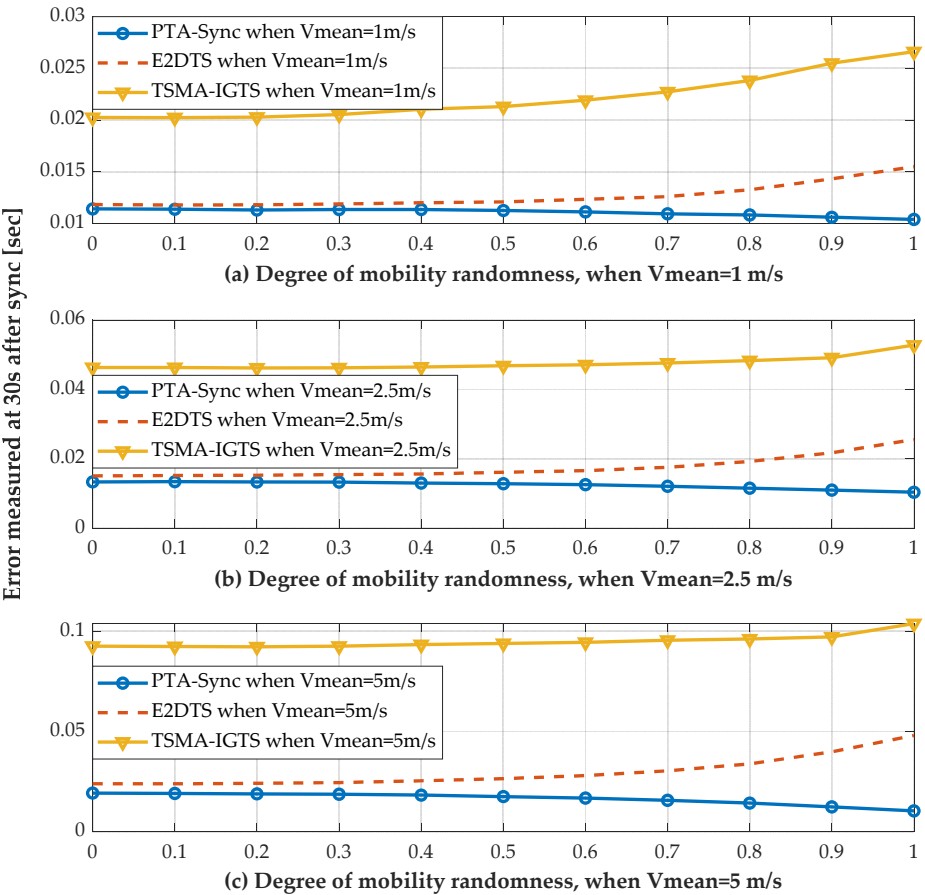

**Figure 7.** Time error measured at 30 s after synchronization versus $\alpha$ when $N\_ex$ = 10, $I\_tx$ = 1.2 s, $Nr$ = 15 km, $T$ = 30 s.

Figure 8 shows the time synchronization error according to the network range. The simulation results were obtained when $Nr$ = 500 m–30 km, $I\_tx$ = 1.2 s, $N\_ex$ = 10, $V\_mean$ = 1 m/s, 2.5 m/s, 5 m/s, $T$ = 30 s, and $\alpha$ = 0.5. As shown in the figure, the time errors of the three synchronization protocols decrease as the network range increases. This is because as the network range increases, the average initial propagation delay with error free value given in the assumption increases, whereas the propagation delay changes with error-prone values are almost constant, regardless of the network range. Therefore, when synchronization is performed, the time error decreases because the influence of uncertain propagation delay changes is reduced compared to the given initial propagation delay. PTA-Sync has the best synchronization performance among the compared protocols, showing an average performance improvement of 79.57% and 45.21% compared to TSMA-IGTS and E2DTS, respectively, when $V\_mean$ = 2.5 m/s.

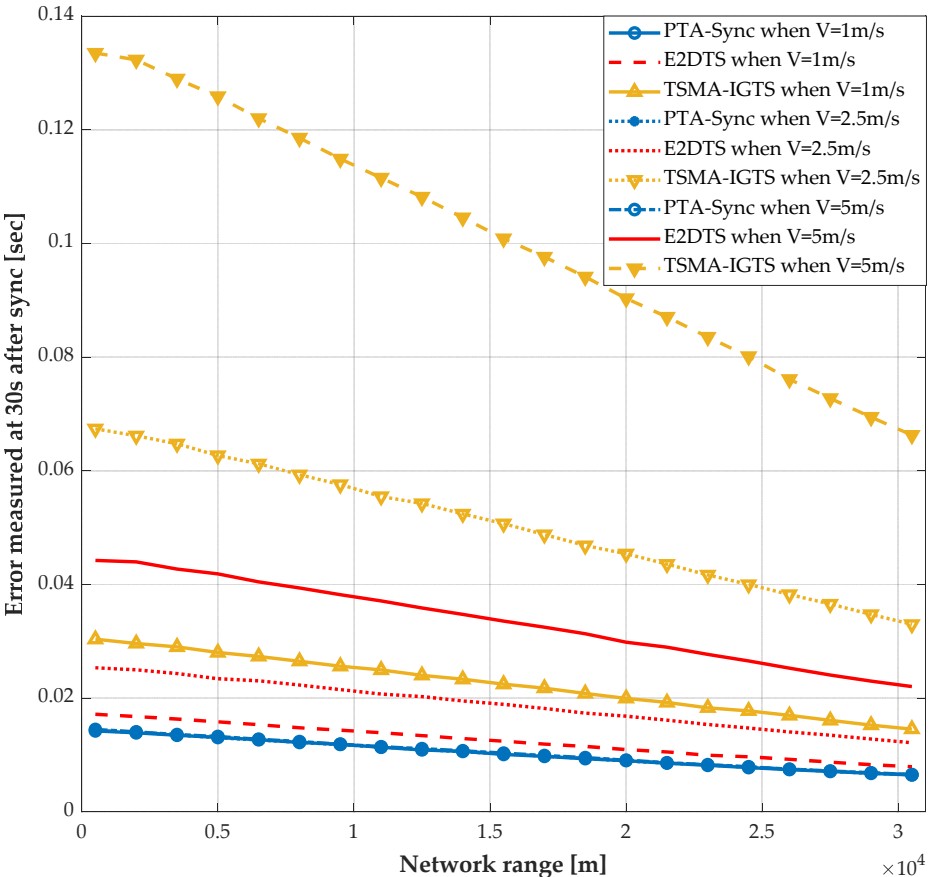

**Figure 8.** Time error measured at 30 s after synchronization versus *Nr* when *V_mean* = 1 m/s, 2.5 m/s, 5 m/s, *N_ex* = 10, *I_tx* = 1.2 s, *T* = 30 s, *α* = 0.5.

## 4. Conclusions

This study proposed PTA-Sync, which adopts the one-way message exchange method for reducing the effect of propagation delay asymmetry as well as minimizing the message overhead for synchronization and delay time. To precisely track the propagation delay changes caused by the node's mobility, the change in propagation delay is continuously estimated using the reference node's position information and the moving node's 3D linear velocity vector, in addition to the transmission time difference and reception time difference information for each packet section transmitted continuously. The simulation results showed that PTA-Sync demonstrated more outstanding performance in relation to one-way communication-based schemes in terms of synchronization accuracy. Further research is verifying the performance of PTA-Sync through real sea experiments by applying the protocols to an underwater communication system. Furthermore, we will conduct research on a more advanced synchronization protocol by adding an algorithm that estimates the uncertainty of underwater propagation delay through a statistical method. The proposed protocol is expected to be practical in underwater acoustic applications with extremely long propagation delay. In addition, by increasing the accuracy of time synchronization, data quality or protocol accuracy can be improved in underwater acoustic networks.

**Author Contributions:** Conceptualization, A.-R.C.; Investigation, A.-R.C.; Validation, A.-R.C.; Writing—original draft preparation, A.-R.C.; Writing—review and editing, Y.C.; Visualization, A.-R.C.; Supervision, Y.C. All authors have read and agreed to the published version of the manuscript.

**Funding:** This work was supported by a grant from the Endowment Project, "Development of Core Technology for Cooperative Navigation of Multiple Marine Robots and Underwater Wireless Cognitive Network", funded by the Korea Research Institute of Ships and Ocean engineering (PES4370).

**Institutional Review Board Statement:** Not applicable.

**Informed Consent Statement:** Not applicable.

**Data Availability Statement:** Not applicable.

**Conflicts of Interest:** The authors declare no conflict of interest.

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
