# Peer review of "PTA-Sync: Packet-Train-Aided Time Synchronization for Underwater Acoustic Applications"

_applsci, doi:10.3390/app13020978_

Round 1

Reviewer 1 Report

Dear Authors,

The paper is well written, but it can be explained in more detail in the under water scenario such as how delay changes with respect to the depth of water.

Reviewer 2 Report

Authors tried to explain the context of UWSN via "Packet-Train-Aided Time Synchronization for Underwater Acoustic Applications". Here, the proposed scheme reduces synchronization error accumulation rate by estimating the propagation delay change. Bus some changes are highly recommended.

1. The manuscript should be complete and it should not be half explained. A lot of points are found in the manuscript where the explanation is missing.

2. Add some latest work of last 3-4 years like a) Priority based data gathering using multiple mobile sinks in cluster based UWSNs for oil pipeline leakage detection b) A range based node localization scheme with hybrid optimization for underwater wireless sensor network c) Hybrid optimization for fault‐tolerant and accurate localization in mobility assisted underwater wireless sensor networks

3. Simulation parameters table is highly desired with specific values and inputs considered here.

4. Result portion is quite weak. Needs to add more result graphs with explanation.

5. Most importantly, I am not able to see the novelty of the work. Mention the novel points at the end of introduction and prove them at the end as well.

6. Include a comparison with similar field approaches at the end of result section. Because only comparing with 2 techniques is not much enough now a days.

Reviewer 3 Report

This paper proposes a clock synchronization algorithm for under water sensor node.  Simulation results show that it is better than other existing underwater synchronization methods. In general, this paper is clearly written and the contribution in the context of underwater synchronization is sufficient for publication.

However, I want to draw the authors' attention to the fact that time synchronization over air and over water may not be that different. The uncertainty of propagation delay can be taken care of by using statistical clock synchronization methods [R1], [R2]. These methods are more advanced than [5]-[8]. In fact, [R1], [R2] can be easily adopted in acoustic propagation. 

This work is based on methods similar to the level of [5]-[8]. This is good enough at this moment as underwater synchronization is relatively less studied compared to that in wireless sensor networks.  A good future direction is to consider statistical methods in underwater synchronization.  This will bring the algorithms for underwater application to the same standard as the state-of-the-art methods in wireless sensor networks.

[R1] ``On Clock Synchronization Algorithms for Wireless Sensor Networks under Unknown Delay," IEEE TVT, 2010.

[R2] ``Joint Time Synchronization and Localization of an unknown node in Wireless Sensor Networks," IEEE TSP 2010.  

Reviewer 4 Report

1. The synchronization performance of the proposed PTA-Sync was compared with that of E2DTS and TSMA-IGTS, thus the complexity analysis of this three may be required.

2. In this article, all performance comparisons are based on individual argument, a simulation of a typical scenario can be added. Besides more typical performance comparisons can be added.

3.  A real sea experiment can further increase the persuasiveness.

4.On page 3,line134,"US" should be "UN", please check the details carefully before final submission.

Reviewer 5 Report

1. Which method is applied for the correction of transmitted data?

2. What is the percentage of error with respect to range?

3. Some more latest papers can be added in the literature survey

4.Rewrite abstract and conclusion

Round 2

Reviewer 2 Report

The paper is not that improved as expected. The simulation parameters are also not realistic. The novelty and inventiveness does not signify about UWSN, it may be reflected in WSN as well. So, I can't recommend for publication in reputed Journal like Applied Sciences.
